# Myomatrix arrays for high-definition muscle recording

Bryce Chung[1†], Muneeb Zia[2†], Kyle A Thomas[3], Jonathan A Michaels[4], Amanda Jacob[1], Andrea Pack[5], Matthew J Williams[3], Kailash Nagapudi[1], Lay Heng Teng[1], Eduardo Arrambide[1], Logan Ouellette[1], Nicole Oey[1], Rhuna Gibbs[1], Philip Anschutz[6], Jiaao Lu[7], Yu Wu[2], Mehrdad Kashefi[4], Tomomichi Oya[4], Rhonda Kersten[4], Alice C Mosberger[8], Sean O'Connell[3], Runming Wang[9], Hugo Marques[10], Ana Rita Mendes[10], Constanze Lenschow[10‡], Gayathri Kondakath[11], Jeong Jun Kim[12], William Olson[12], Kiara N Quinn[13], Pierce Perkins[13], Graziana Gatto[14§], Ayesha Thanawalla[14], Susan Coltman[15], Taegyo Kim[16], Trevor Smith[16], Ben Binder-Markey[17], Martin Zaback[18], Christopher K Thompson[18], Simon Giszter[16], Abigail Person[15,19], Martyn Goulding[14], Eiman Azim[14], Nitish Thakor[13], Daniel O'Connor[12], Barry Trimmer[11], Susana Q Lima[10], Megan R Carey[10], Chethan Pandarinath[9], Rui M Costa[8], J Andrew Pruszynski[4], Muhannad Bakir[2], Samuel J Sober[1*]

*For correspondence: samuel.j.sober@emory.edu

†These authors contributed equally to this work

Present address: ‡Institute of Biology, Otto-von-Guericke University, Magdeburg, Germany; §Department of Neurology, University Hospital of Cologne, Cologne, Germany

[1]Department of Biology, Emory University, Atlanta, United States; [2]School of Electrical and Computer Engineering, Georgia Institute of Technology, Atlanta, United States; [3]Graduate Program in Biomedical Engineering at Emory University and Georgia Tech, Atlanta, United States; [4]Department of Physiology and Pharmacology, Western University, London, Canada; [5]Neuroscience Graduate Program, Emory University, Atlanta, United States; [6]Graduate Program in BioEngineering, Georgia Tech, Atlanta, United States; [7]Graduate Program in Electrical and Computer Engineering, Georgia Tech, Atlanta, United States; [8]Zuckerman Mind Brain Behavior Institute at Columbia University, New York, United States; [9]Department of Biomedical Engineering at Emory University and Georgia Tech, Atlanta, United States; [10]Champalimaud Neuroscience Programme, Champalimaud Foundation, Lisbon, Portugal; [11]Department of Biology, Tufts University, Medford, United States; [12]Solomon H. Snyder Department of Neuroscience, Johns Hopkins School of Medicine, Baltimore, United States; [13]Departments of Biomedical Engineering and Neurology, Johns Hopkins School of Medicine, Baltimore, United States; [14]Salk Institute for Biological Studies, La Jolla, United States; [15]Department of Physiology and Biophysics, University of Colorado Anschutz Medical Campus, Aurora, United States; [16]Department of Neurobiology & Anatomy, Drexel University, College of Medicine, Philadelphia, United States; [17]Department of Physical Therapy and Rehabilitation Sciences, Drexel University College of Nursing and Health Professions, Philadelphia, United States; [18]Department of Health and Rehabilitation Sciences, Temple University, Philadelphia, United States; [19]Allen Institute, Seattle, United States

**Abstract** Neurons coordinate their activity to produce an astonishing variety of motor behaviors. Our present understanding of motor control has grown rapidly thanks to new methods for recording and analyzing populations of many individual neurons over time. In contrast, current methods for recording the nervous system's actual motor output – the activation of muscle fibers by motor neurons – typically cannot detect the individual electrical events produced by muscle fibers during

natural behaviors and scale poorly across species and muscle groups. Here we present a novel class of electrode devices ('Myomatrix arrays') that record muscle activity at unprecedented resolution across muscles and behaviors. High-density, flexible electrode arrays allow for stable recordings from the muscle fibers activated by a single motor neuron, called a 'motor unit,' during natural behaviors in many species, including mice, rats, primates, songbirds, frogs, and insects. This technology therefore allows the nervous system's motor output to be monitored in unprecedented detail during complex behaviors across species and muscle morphologies. We anticipate that this technology will allow rapid advances in understanding the neural control of behavior and identifying pathologies of the motor system.

## eLife assessment

This **important** paper reports major technical advances for in vivo intramuscular electrical recording from multiple motor units in behaving animals. The paper includes **compelling** demonstrations of the efficacy of this new technique in multiple animal species. This new muscle recording method has the potential to provide new insight into a wide range of questions in motor neuroscience.

## Introduction

Recent decades have seen tremendous advances in our understanding of the physiological mechanisms by which the brain controls complex motor behaviors. Critical to these advances have been tools to record neural activity at scale (*Steinmetz et al., 2018*; *Urai et al., 2022*), which, when combined with novel algorithms for behavioral tracking (*Machado et al., 2015*; *Berman et al., 2014*; *Pereira et al., 2019*; *Mathis et al., 2020*; *Wiltschko et al., 2020*), can reveal how neural activity shapes behavior (*Hernández et al., 2022*; *Vyas et al., 2020*). In contrast, current methods for observing the nervous system's motor output lag far behind neural recording technologies. The nervous system's control of skeletal motor output is ultimately mediated by 'motor units,' each of which consists of a single motor neuron and the muscle fibers it activates, producing motor unit action potentials (*Figure 1a*) that generate muscle force to produce movement (*Manuel et al., 2019*). Because each action potential in a motor neuron reliably evokes a single spike in its target muscle fibers, action potentials recorded from muscle provide a high-resolution readout of motor neuron activity in the spinal cord and brainstem. However, our understanding of motor unit activity during natural behaviors is rudimentary due to the difficulty of recording spike trains from motor unit populations.

Traditional methods for recording muscle fiber activity via electromyography (EMG) include fine wires inserted into muscles and electrode arrays placed on the surface of the skin (*Loeb and Gans, 1986*). These methods can resolve the activity of individual motor units in only a limited range of settings. First, to prevent measurement artifacts, traditional EMG methods require that a subject's movements be highly restricted, typically in 'isometric' force tasks where subjects contract their muscles without moving their bodies (*Bräcklein et al., 2022*; *Farina and Holobar, 2016*; *Marshall et al., 2022*; *Negro et al., 2016*). Moreover, fine-wire electrodes typically cannot detect single motor unit activity in small muscles, including the muscles of widely used model systems such as mice or songbirds (*Pearson et al., 2005*; *Srivastava et al., 2015*; *Pack et al., 2023*), and surface electrode arrays are poorly tolerated by freely behaving animal subjects. These limitations have impeded our understanding of fundamental questions in motor control, including how the nervous system coordinates populations of motor units to produce skilled movements, how this coordination degrades in pathological states, and how motor unit activity is remapped when animals learn new tasks or adapt to changes in the environment.

Here, we present a novel approach (*Figure 1*) to recording populations of individual motor units from many different muscle groups and species during natural behaviors. Flexible multielectrode ('Myomatrix') arrays were developed to achieve the following goals:

1. Record muscle activity at motor unit resolution
2. Record motor units during active movements
3. Record from a wide range of muscle groups, species, and behaviors
4. Record stably over time and with minimal movement artifact

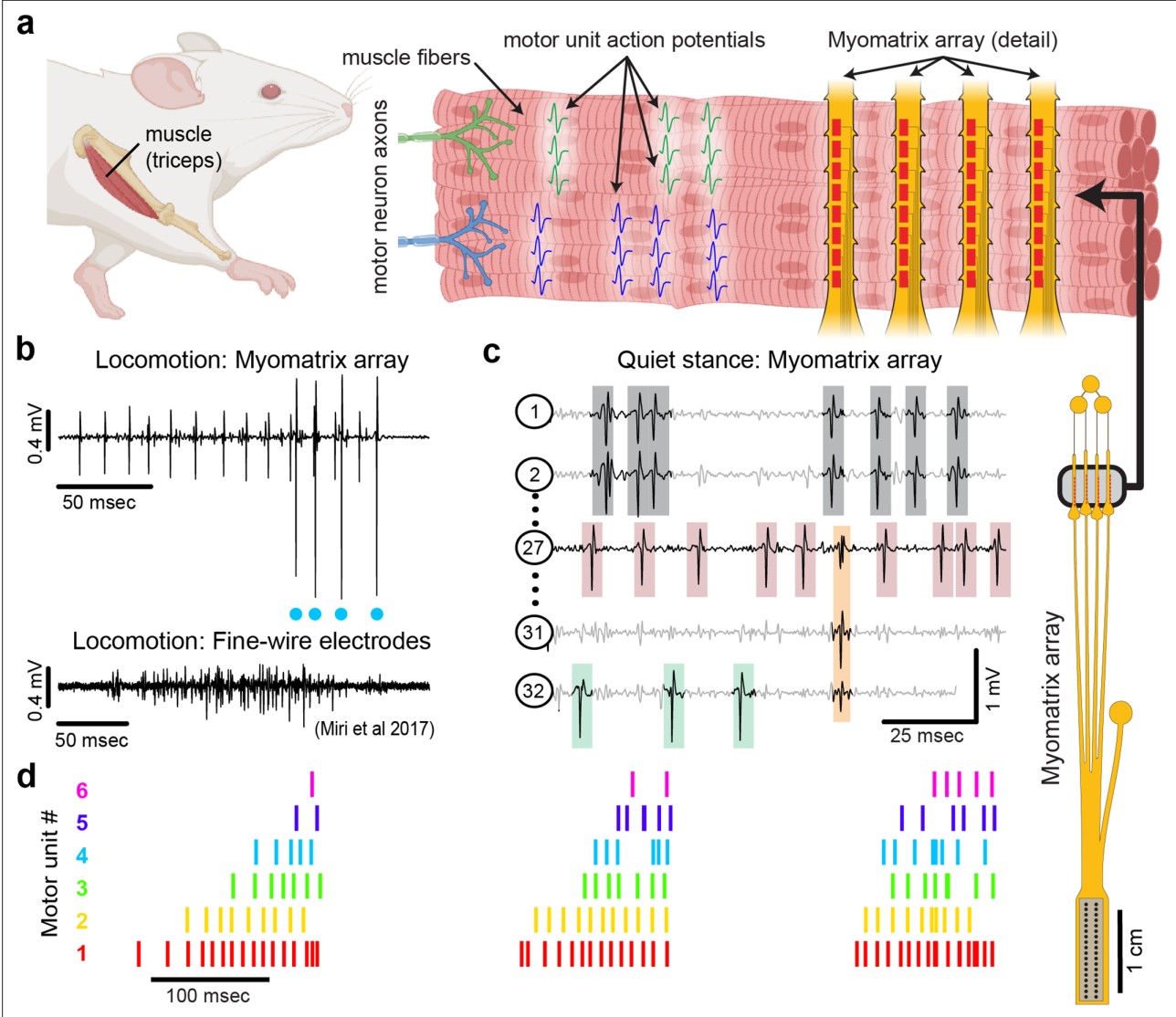

**Figure 1.** Myomatrix arrays record muscle activity at motor unit resolution. (**a**) The nervous system controls behavior via motor units, each consisting of a single motor neuron and the muscle fibers it innervates. Each motor neuron's spiking evokes motor unit action potentials in the corresponding muscle fibers. Myomatrix arrays (right) bearing 32 electrode contacts on a flexible substrate (**Figure 1—figure supplement 1**) can be targeted to one or more muscles and yield high-resolution recordings of motor activity during free behavior. Motor neurons, muscle fibers, and electrode arrays are not shown to scale. (**b, c**) Example recordings from the right triceps muscle of a freely behaving mouse. (**b**) Top: bipolar Myomatrix recording from the mouse triceps during locomotion. Blue dots indicate the spike times of one motor unit isolated from the data using a spike-sorting method based on principal components analysis (PCA) (**Figure 1—figure supplement 2a–d**). Bottom: example data (from **Miri et al., 2017**, used with permission) from traditional fine-wire EMG recording of triceps activity during locomotion. Applying the PCA-based spike-sorting method to the fine-wire data did not isolate any individual motor units. (**c**) Unipolar Myomatrix recording during quiet stance. Colored boxes illustrate motor unit action potentials from four identified units. Spike waveforms from some units, including those highlighted with gray and orange boxes, appear on multiple electrode channels, requiring the use of a multichannel spike-sorting algorithm (see **Figure 1—figure supplement 2e–h**). (**d**) Spiking pattern (tick marks) of six individual motor units recorded simultaneously during locomotion on a treadmill. The three bursts of motor unit action potentials correspond to triceps activity during three stride cycles. Motor unit 4 (cyan) is the same motor unit represented by cyan dots in (**b**). The other motor units in this recording, including the smaller amplitude units at top in (**b**), were isolated using Kilosort but could not be isolated with the PCA-based method applied to data from only the single recording channel shown (**b**).

The online version of this article includes the following figure supplement(s) for figure 1:

**Figure supplement 1.** Myomatrix fabrication, design variations, and implantation.

**Figure supplement 2.** Spike sorting.

**Figure supplement 3.** Longevity of Myomatrix recordings.

To achieve these goals, we developed a variety of array configurations for use across species and muscle groups. Voltage waveforms from individual motor units (*Figure 1b and c*) can be readily extracted from the resulting data using a range of spike-sorting algorithms, including methods developed to identify the waveforms of individual neurons in high-density arrays (*Pachitariu et al., 2023*; *Sober et al., 2008*). Below, we show how Myomatrix arrays provide high-resolution measurements of motor unit activation in a variety of species and muscle groups including forelimb, hindlimb, orofacial, pelvic, vocal, and respiratory muscles.

## Results and discussion

We developed methods to fabricate flexible, high-density EMG ('Myomatrix') arrays, as detailed in the 'Methods' and schematized in *Figure 1—figure supplement 1*. We selected polyimide as a substrate material due to its strength and flexibility and the ease with which we could define electrode contacts, suture holes, and other sub-millimeter features that facilitate implantation and recording stability (*Figure 1—figure supplement 1a–e*). Moreover, simple modifications to the fabrication pipeline allowed us to rapidly design, test, and refine different array morphologies targeting a range of muscle sizes, shapes, and anatomical locations (*Figure 1—figure supplement 1c, f, and g*).

### Myomatrix arrays record muscle activity at motor unit resolution

Myomatrix arrays robustly record the activity of individual motor units in freely behaving mice. Arrays specialized for mouse forelimb muscles include four thin 'threads' (8 electrodes per thread, 32 electrodes total) equipped with suture holes, flexible barbs, and other features to secure the device within or onto a muscle (*Figure 1a*, *Figure 1—figure supplement 1c–e and h*). These devices yielded well-isolated waveforms from individual motor units (*Figure 1b*, top), which were identified using open-source spike-sorting tools (*Pachitariu et al., 2023*; *Sober et al., 2008*). As detailed in *Figure 1—figure supplement 2a–d*, in some cases the spike times of individual motor units (*Figure 1b*, cyan dots) can be isolated from an individual electrode channel with simple spike-sorting approaches including single-channel waveform clustering (*Sober et al., 2008*). In other cases, waveforms from individual motor units appeared on multiple electrode channels (*Figure 1c*), allowing – and in many cases necessitating – more advanced spike-sorting approaches that leverage information from multiple channels to identify larger numbers of units and resolve overlapping spike waveforms (*Pachitariu et al., 2023*), as detailed in *Figure 1—figure supplement 2e–h*. These methods allow the user to record simultaneously from ensembles of single motor units (*Figure 1c and d*) in freely behaving animals, even from small muscles including the lateral head of the triceps muscle in mice (approximately 9 mm in length with a mass of 0.02 g; *Mathewson et al., 2012*). Myomatrix recordings isolated single motor units for extended periods (greater than 2 mo, *Figure 1—figure supplement 3e*), although highest unit yield was typically observed in the first 1–2 wk after chronic implantation. Because recording sessions from individual animals were often separated by several days during which animals were disconnected from data collection equipment, we are unable to assess based on the present data whether the same motor units can be recorded over multiple days.

### Myomatrix arrays record motor units during active movements

Myomatrix arrays outperform traditional fine-wire electrodes in mice by reliably recording isolated single units in behaving animals. First, Myomatrix arrays isolate the activity of multiple individual motor units during freely moving behavior (*Figure 1c and d*). In contrast, wire electrodes typically cannot resolve individual motor units during muscle lengthening and shorting, as occurs in naturalistic movements such as locomotion (*Miri et al., 2017*; *Tysseling et al., 2013*). *Figure 1b* illustrates a comparison between Myomatrix (top) and fine-wire (bottom) data recorded during locomotion in the mouse triceps. Spike sorting identified well-isolated motor unit spikes in the Myomatrix data (*Figure 1b*, top, cyan dots) but failed to extract any isolated motor units in the fine-wire data (*Figure 1—figure supplement 2a and b*). Similarly, while Myomatrix recordings robustly isolated motor units from a songbird vocal muscle, fine-wire EMG electrodes applied to the same muscle did not yield isolatable units (*Figure 1—figure supplement 2c and d*). This lack of resolution, which is typical of fine-wire EMG, severely limits access to motor unit activity during active behavior, although wire electrodes injected through the skin can provide excellent motor unit isolation during quiet stance in mice (*Ritter et al.,*

*2014*). Second, because wire-based EMG requires inserting an additional wire for each additional electrode contact, only a single pair of wires (providing a single bipolar recording channel, *Figure 1b*, bottom) can be inserted into an individual mouse muscle in most cases (*Miri et al., 2017*; *Tysseling et al., 2013*; *Tysseling et al., 2017*). In contrast, at least four Myomatrix 'threads' (*Figure 1a*), bearing a total of 32 electrodes, can be inserted into one muscle (*Figure 1c* shows 5 of 32 channels recorded simultaneously from mouse triceps), greatly expanding the number of recording channels within a single muscle. Single motor units were routinely isolated during mouse locomotion in our Myomatrix recordings (*Figure 1*), but never in the fine-wire datasets from *Miri et al., 2017* we reanalyzed or, to our knowledge, in any prior study. Moreover, in multiunit recordings, Myomatrix arrays have significantly higher signal-to-noise ratios than fine-wire EMG arrays (*Figure 1—figure supplement 3*). Myomatrix arrays therefore far exceed the performance of wire electrodes in mice in terms of both the quality of recordings and the number of channels that can be recorded simultaneously from one muscle.

## Myomatrix arrays record from a wide range of muscle groups, species, and behaviors

Myomatrix arrays provide high-resolution EMG recordings across muscle targets and experimental preparations (*Figure 2*). Beyond the locomotor and postural signals shown in *Figure 1*, Myomatrix recordings from mouse triceps also provided single-unit EMG data during a head-fixed reaching task (*Figure 2a*). In addition to recording single motor units during these voluntary behaviors, Myomatrix arrays also allow high-resolution recordings from other muscle groups during reflex-evoked muscle activity. *Figure 2b* shows single motor unit EMG data recorded from the superficial masseter (jaw) muscle when reflexive muscle contraction was obtained by passively displacing the jaw of an anesthetized mouse. To extend these methods across species, we collected Myomatrix recordings from muscles of the rat forelimb, obtaining isolated motor units from the triceps during locomotion (*Figure 2c*) and a digit-flexing muscle in the lower forearm during head-free reaching (*Figure 2d*). Myomatrix arrays can furthermore isolate motor unit waveforms evoked by direct optogenetic stimulation of spinal motor neurons. *Figure 2e* shows recordings of light-evoked spikes in the mouse bulbospongiosus muscle (BSM, a pelvic muscle that wraps around the base of the penis in male mice), demonstrating that optogenetic stimulation of the spinal cord evokes spiking in single motor units with millisecond-scale timing jitter (*Figure 2e*, center) and with latencies (*Figure 2e*, right) consistent with the latencies of recordings obtained with fine-wire electrodes (*Lenschow et al., 2022*). Beyond rodents, simple modifications of the basic electrode array design (*Figure 1—figure supplement 1f*) allowed us to obtain high-resolution recordings from hindlimb muscles in cats (*Figure 2f*), vocal and respiratory muscles in songbirds (*Figure 2g and h*, see also *Zia et al., 2020*), body wall muscles in moth larvae (*Figure 2i*), and leg muscles in frogs (*Figure 2j*).

In addition to isolating individual motor units, Myomatrix arrays provide stable multiunit recordings of comparable or superior quality to conventional fine-wire EMG. Although single-unit recordings are essential to identify individual motor neurons' contributions to muscle activity (*Sober et al., 2018*; *Srivastava et al., 2017*), for other lines of inquiry a multiunit signal is preferred as it reflects the combined activity of many motor units within a single muscle. Although individual Myomatrix channels are often dominated by spike waveforms from one or a small number of motor units (*Figure 1b*), other channels reflect the combined activity of multiple motor units as typically observed in fine-wire EMG recordings (*Quinlan et al., 2017*). As shown in *Figure 1—figure supplement 3a and b*, these multiunit Myomatrix signals are stable over multiple weeks of recordings, similar to the maximum recording longevity reported for wire-based systems in mice and exceeding the 1–2 wk of recording more typically obtained with wire electrodes in mice (*Miri et al., 2017*; *Tysseling et al., 2013*; *Tysseling et al., 2017*), and with significantly greater recording quality than that obtained from wire electrodes at comparable post-implantation timepoints (*Figure 1—figure supplement 3d*).

To record from larger muscles than those described above, we also created designs targeting the forelimb and shoulder muscles of rhesus macaques (*Figure 3*). Although fine-wire electrodes have been used to isolate individual motor units in both humans and monkeys (*Loeb and Gans, 1986*; *Marshall et al., 2021*), and skin-surface electrode arrays robustly record motor unit populations in human subjects (*Bräcklein et al., 2022*; *Farina and Merletti, 2000*), this resolution is limited to isometric tasks – that is, muscle contraction without movement – due to the sensitivity of both

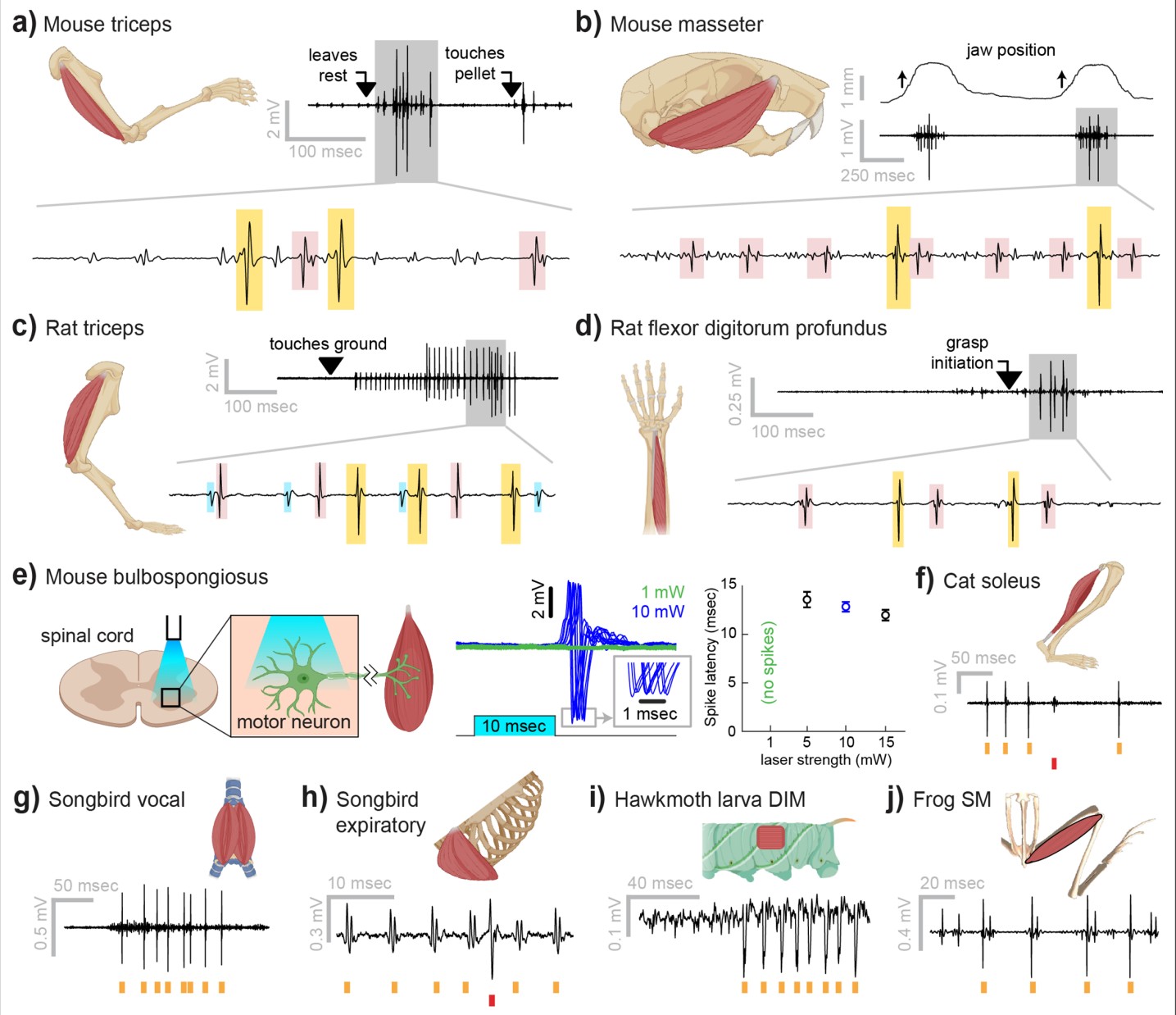

**Figure 2.** Myomatrix recordings across muscles and species. (**a**) Example recording from mouse triceps during a head-fixed pellet reaching task. Arrows at top indicate the approximate time that the animal's paw leaves a rest position and first contacts the target. Bottom: colored boxes highlight motor unit action potentials identified using a multichannel spike sorting algorithm (**Pachitariu et al., 2023**). Different box colors on the same voltage trace indicate distinct motor units. (**b**) Recordings from the mouse superficial masseter muscle were obtained in anesthetized, head-fixed mice when passive mandible displacement evoked reflexive muscle contractions. Top trace shows the lateral component of jaw displacement, with arrows indicating the direction and approximate time of displacement onset. (**c**) In a recording from rat triceps during head-free locomotion, the arrowhead indicates the time that the mouse's paw touched the treadmill surface, marking the beginning of the stance phase. (**d**) Recording from the rat flexor digitorum profundus muscle during a pellet reaching task, arrow indicates the time of grasp initiation. (**e**) Myomatrix recording of motor unit activity in the mouse bulbospongiosus muscle evoked by optical stimulation of spinal motor neurons, producing motor unit spikes at latencies between 10 and 15 ms, consistent with results obtained from traditional fine-wire electrodes in mice (**Lenschow et al., 2022**). (**f–j**) Recordings from the cat soleus (**f**) during sensory nerve stimulation, songbird vocal (ventral syringeal) muscle (**g**) and expiratory muscle (**h**) during quiet respiration, hawkmoth larva dorsal internal medial (DIM) muscle (**i**) during fictive locomotion, and bullfrog semimembranosus (SM) muscle (**j**) in response to cutaneous (foot) stimulation. Spike times from individual motor units are indicated by colored tick marks under each voltage trace in (**f–j**). Recordings shown in panels (**a, c, g, h, i**, and **j**) were collected using bipolar amplification, data in panels (**b, d, e**, and **f**) were collected using unipolar recording. See 'Methods' for details of each experimental preparation.

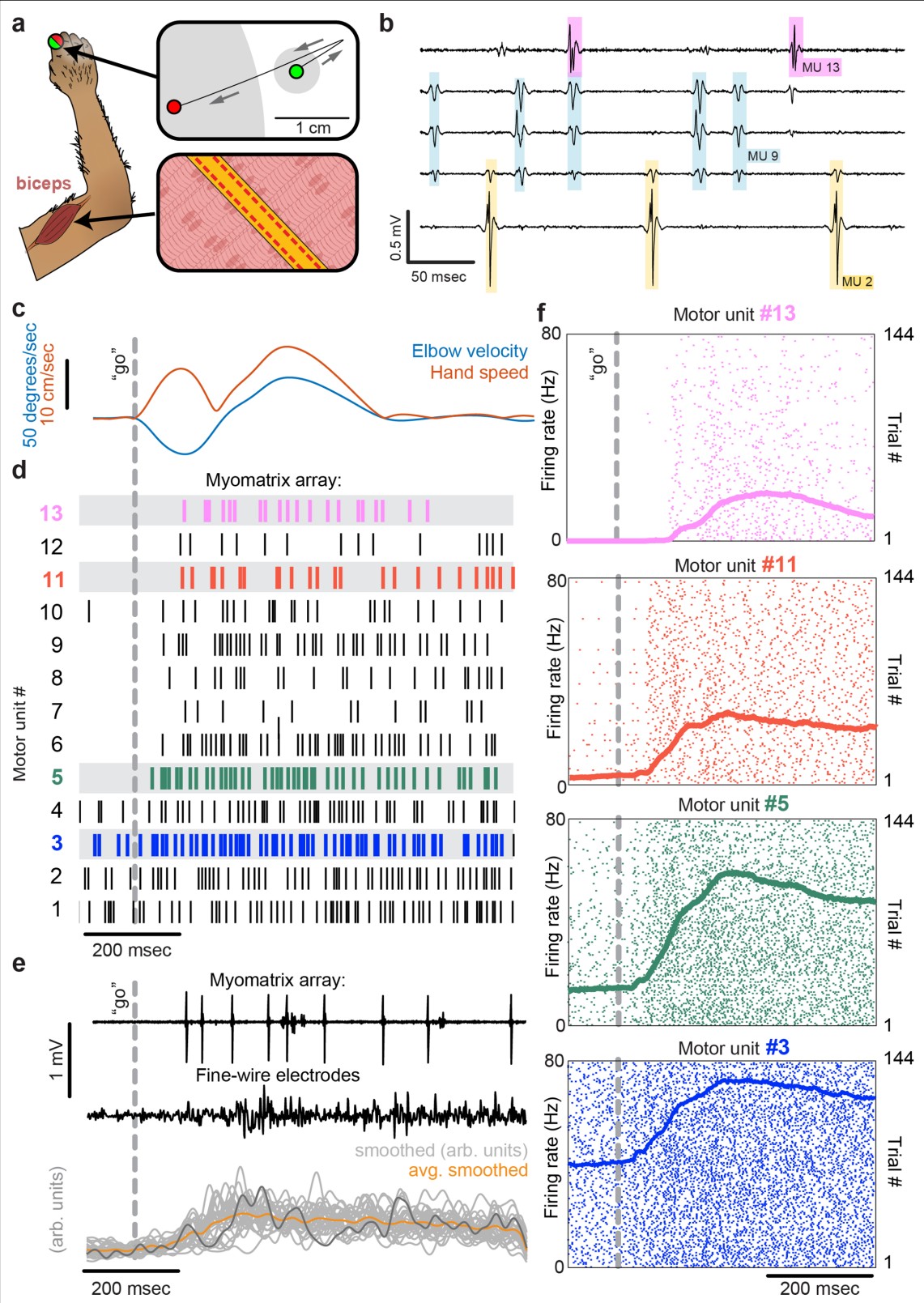

**Figure 3.** Motor unit recordings during active movement in primates. (**a**) An injectable version of the Myomatrix array (*Figure 1—figure supplement 1g*) was inserted percutaneously (*Figure 1—figure supplement 1i*) into the right biceps of a rhesus macaque performing a cued reaching task. Green and red dots: reach start and endpoints, respectively; gray regions: start and target zones. (**b**) Recording from 5 of 32 unipolar channels showing spikes from three individual motor units isolated from the multichannel recording (*Figure 1—figure supplement 2*). (**c**) At trial onset (dotted line), a sudden

*Figure 3 continued on next page*

*Figure 3 continued*

force perturbation extends the elbow, signaling the animal to reach to the target. (**d**) Spike times (tick marks) from 13 simultaneously recorded motor units. (**e**) Example voltage data from a Myomatrix array (top) and traditional fine-wire EMG (middle, bottom) collected from the same biceps muscle in the same animal performing the same task, but in a separate recording session. Gray traces (bottom) show smoothed EMG data from the fine-wire electrodes in all trials, orange trace shows trial-averaged smoothed fine-wire EMG, and dark gray trace represents smoothed data from the example fine-wire trace shown above it. (**f**) Spike times of four motor units (of the 13 shown in **d**) recorded simultaneously over 144 trials.

fine-wire and surface array electrodes to electrical artifacts caused by body movement. For ease of insertion into larger muscles, we modified the 'thread' design used in our mouse arrays so that each Myomatrix array could be loaded into a standard hypodermic syringe and injected into the muscle (*Figure 1—figure supplement 1g and i*), inspired by earlier work highlighting the performance of injectable arrays in primates (*Loeb and Gans, 1986*; *Farina et al., 2008*; *Muceli et al., 2015*). As shown in *Figure 3a–d*, injectable Myomatrix arrays yielded motor unit recordings during arm movements. Tick marks in *Figure 3d* show the activity of 13 motor units recorded simultaneously during a single trial in which a monkey was cued (by a force perturbation which causes an extension of the elbow joint) to reach to a target. In contrast to the ensemble of motor units recorded with a Myomatrix probe, conventional fine-wire EMGs inserted into the same muscle (in a separate recording session) yielded only a single trace reflecting the activity of an unknown number of motor units (*Figure 3e*, middle). Moreover, although fine-wire EMG signals varied across trials (*Figure 3e*, bottom), the lack of motor unit resolution makes it impossible to assess how individual motor units vary (and co-vary) across trials. In contrast, Myomatrix recordings provide spiking resolution of multiple individual motor units across many trials, as illustrated in *Figure 3f*.

## Myomatrix arrays record stably over time and with minimal movement artifact

Myomatrix arrays provide EMG recordings that are resistant to movement artifacts and stable over time. In recordings from rodents during locomotion (*Figures 1b and 2c*), we did not observe voltage artifacts at any point in the stride cycle (e.g. when the paw of the recorded limb first touches the ground in each stride cycle; *Figure 2c*, black arrowhead). Movement artifacts were similarly absent during active arm movements in monkeys (peak fingertip speeds ~15 cm/s; peak elbow angle velocity ~40°/s, *Figure 3b and e*), during passive jaw displacement in anesthetized mice (*Figure 2b*), or in the other anesthetized preparations shown in *Figure 2*. Individual motor units could typically be isolated for the entire duration of an acute or chronic recording session. In triceps recordings during locomotion in rodents (*Figures 1 and 2c*), isolated motor units were recorded for up to 4791 stride cycles (mice) or 491 stride cycles (rats) during continuous recording sessions lasting 10–60 min. Myomatrix recordings in behaving nonhuman primates were similarly long-lived, as in the dataset shown in *Figure 3*, where single-unit isolation was maintained across 1292 reaching trials collected over 97 min. In each of these datasets, the duration of single-unit EMG recording was limited by the willingness of the animal to continue performing the behavior, rather than a loss of signal isolation. Recordings in acute preparations were similarly stable. For example, the songbird dataset shown in *Figure 2g* includes single-unit data from 8101 respiratory cycles collected over 74 min, and, like the other acute recordings shown in *Figure 2*, recordings were ended by the experimenter rather than because of a loss of signal from individual motor units.

The diversity of applications presented here demonstrates that Myomatrix arrays can obtain high-resolution EMG recordings across muscle groups, species, and experimental conditions, including spontaneous behavior, reflexive movements, and stimulation-evoked muscle contractions. Although this resolution has previously been achieved in moving subjects by directly recording from motor neuron cell bodies in vertebrates (*Hoffer et al., 1981*; *Robinson, 1970*; *Hyngstrom et al., 2007*) and using fine-wire electrodes in moving insects (*Pflüger and Burrows, 1978*; *Putney et al., 2023*), both methods are extremely challenging and can only target a small subset of species and motor unit populations. Exploring additional muscle groups and model systems with Myomatrix arrays will allow new lines of investigation into how the nervous system executes skilled behaviors and coordinates populations of motor units both within and across individual muscles. These approaches will be particularly valuable in muscles in which each motor neuron controls a very small number of muscle fibers, allowing fine control of oculomotor muscles in mammals as well as vocal muscles in songbirds

(*Figure 2g*), in which most individual motor neurons innervate only 1–3 muscle fibers (*Adam et al., 2021*). Of further interest will be combining high-resolution EMG with precise measurement of muscle length and force output to untangle the complex relationship between neural control, body kinematics, and muscle force that characterizes dynamic motor behavior. Similarly, combining Myomatrix recordings with high-density brain recordings or targeted manipulations of neural activity can reveal how central circuits shape and reshape motor activity and – in contrast to the multiunit signals typically obtained from traditional EMG in animals – reveal how neural dynamics in cortical, subcortical, and spinal circuits shape the spiking patterns of individual motor neurons.

Applying Myomatrix technology to human motor unit recordings, particularly by using the minimally invasive injectable designs shown in *Figure 3* and *Figure 1—figure supplement 1g and i*, will create novel opportunities to diagnose motor pathologies and quantify the effects of therapeutic interventions in restoring motor function. Moreover, because Myomatrix arrays are far more flexible than the rigid needles commonly used to record clinical EMG, our technology might significantly reduce the risk and discomfort of such procedures while also greatly increasing the accuracy with which human motor function can be quantified. This expansion of access to high-resolution EMG signals – across muscle groups, species, and behaviors – is the chief impact of the Myomatrix project.

## Methods
### Myomatrix array fabrication
The microfabrication process (schematized in *Figure 1—figure supplement 1a*) consists of depositing and patterning a series of polymer (polyimide) and metal (gold) layers using a combination of spin coating, photolithography, etching, and evaporation processes, as described previously (*Zia et al., 2020*; *Lu et al., 2022*; *Zia et al., 2018*). These methods allow very fine pitch escape routing (<10 μm spacing between the thin 'escape' traces connecting electrode contacts to the connector), spatial alignment between the multiple layers of polyimide and gold that constitute each device, and precise definition of 'via' pathways that connect different layers of the device. Once all the metal and polyimide layers have been deposited and patterned on carrier wafers, the gold EMG recording electrode sites are formed by removing the top polyimide layer over each electrode site using reactive ion etching process ($O_2$ and $SF_6$ plasma, 5:1 ratio). Electrode sites are then coated with a conductive polymer, PEDOT:PSS (Poly(3,4-ethylenedioxythiophene)-poly(styrene-sulfonate)), (*Cui and Martin, 2003*; *Dijk et al., 2020*; *Rossetti et al., 2019*) to reduce the electrode impedance (*Ludwig et al., 2011*). PEDOT:PSS was deposited on the electrode contacts to a thickness of 100 nm using spin coating, resulting in final electrode impedances of 5 kOhm or less (100 × 200 um electrode sites). Once all layers have been deposited on the carrier wafer, the wafer is transferred to an Optec Femtosecond laser system, which is used to cut the electrode arrays into the shape/pattern needed based on the target muscle group and animal species. The final device thickness was ~40 μm for the injectable (primate forelimb) design and ~20 μm for all other design variants. The final fabrication step is bonding a high-density connector (Omnetics, Inc) to the surface of the electrode array using a Lambda flip-chip bonder (Finetech, Inc). This fabrication pipeline allows the rapid development and refinement of multiple array designs (*Figure 1—figure supplement 1c–g*).

### Myomatrix array implantation
For chronic EMG recording in mice and rats (*Figures 1 and 2a, c, and d*), arrays such as those shown in *Figure 1—figure supplement 1c–f* were implanted by first making a midline incision (approximately 10 mm length) in the scalp of an anesthetized animal. The array's connector was then secured to the skull using dental cement (in some cases along with a headplate for later head-fixed chronic recordings), and the electrode array threads were routed subcutaneously to a location near the target muscle or muscles (*Figure 1—figure supplement 1h*). In some electrode array designs, subcutaneous routing is facilitated with 'pull-through tabs' that can be grasped with a forceps to pull multiple threads into position simultaneously. For some anatomical targets, a small additional incision was made to allow surgical access to individual muscles (e.g. a 2–5 mm incision near the elbow to facilitate implantation into the biceps and/or triceps muscles). Once each thread has been routed subcutaneously and positioned near its target, any pull-through tabs are cut off with surgical scissors and discarded. Each thread can then either be sutured to the surface of the thin sheet of elastic tissue

that surrounds muscles ('epimysial attachment') or inserted into the muscle using a suture needle ('intramuscular implantation'). For epimysial attachment, each electrode thread is simply sutured to the surface of each muscle (with electrode contacts facing the muscle tissue, suture sizes ranging from 6-0 to 11–0) in one of the proximal suture holes (located on the depth-restrictor tabs) and one of the distal suture holes. For intramuscular implantation (*Figure 1—figure supplement 1h*), a suture (size 6-0 to 11-0 depending on anatomical target) is tied to the distal-most suture hole. The needle is then passed through the target muscle and used to pull the attached array thread into the muscle. In some designs, a 'depth-restrictor tab' (*Figure 1—figure supplement 1d*) prevents the thread from being pulled any further into the muscle, thereby limiting the depth at which the electrodes are positioned within the target muscle. The array is then secured within the muscle by the passive action of the flexible polyimide 'barbs' lining each thread and/or by adding additional sutures to the proximal and distal suture holes.

Acute recording in small animals (including rodents, songbirds, cats, frogs, and caterpillars; *Figure 2b and e–j*) used the same arrays as chronic recordings. However, for both epimysial and intramuscular acute recordings, the Myomatrix array traces were simply placed on or within the target muscle after the muscle was exposed via an incision in the overlying skin of the anesthetized animal (rather than routed subcutaneously from the skull as in chronic applications).

For acute recordings in nonhuman primates, prior to recording, the 'tail' of the injectable array (*Figure 1—figure supplement 1g*) was loaded into a sterile 23-gauge cannula (1″ long) until fully seated. The upper half of the cannula bevel, where contact is made with the electrode, was laser-blunted to prevent breakage of the tail (*Muceli et al., 2015*). During insertion (*Figure 1—figure supplement 1i*), the tail was bent over the top of the cannula and held tightly, and the electrode was inserted parallel to bicep brachii long head muscle fibers at an angle of ~45° to the skin. Once the cannula was fully inserted, the tail was released, and the cannula slowly removed. After recording, the electrode and tail were slowly pulled out of the muscle together. Insertion and removal of injectable Myomatrix devices appeared to be comparable or superior to traditional fine-wire EMG electrodes (in which a 'hook' is formed by bending back the uninsulated tip of the recording wire) in terms of ease of injection, ease of removal of both the cannula and the array itself, and animal comfort. Moreover, in over 100 Myomatrix injections performed in rhesus macaques, there were zero cases in which Myomatrix arrays broke such that electrode material was left behind in the recorded muscle, representing a substantial improvement over traditional fine-wire approaches, in which breakage of the bent wire tip regularly occurs (*Loeb and Gans, 1986*).

For all Myomatrix array designs, a digitizing, multiplexing headstage (Intan, Inc) was plugged into the connector, which was cemented onto the skull for chronic applications and attached to data collection devices via a flexible tether, allowing EMG signals to be collected during behavior. By switching out different headstages, data from the same 32 electrode channels on each Myomatrix array could be recorded either as 32 unipolar channels or as 16 bipolar channels, where each bipolar signal is computed by subtracting the signals from physically adjacent electrode contacts.

## Data analysis: Spike sorting

Motor unit action potential waveforms from individual motor units were identified with analysis methods previously used to sort spikes from neural data. In all cases, Myomatrix signals (sampling rate 30 or 40 kHz) were first band-passed between 350 and 7000 Hz. When the voltage trace from a single Myomatrix channel is dominated by a single high-amplitude action potential waveform (as in *Figure 1b*), single units can be isolated using principal components analysis (PCA) to detect clusters of similar waveforms, as described previously (*Sober et al., 2008*). As detailed in *Figure 1—figure supplement 2a–d*, this method provides a simple quantitative measure of motor unit isolation by quantifying the overlap between clusters of spike waveforms in the space of the first two principal components.

In other cases (as in *Figure 1c*), the spikes of individual motor units appear on multiple channels and/or overlap with each other in time, requiring a more sophisticated spike-sorting approach to identifying the firing times of individual motor units. We therefore adapted Kilosort version 2.5 (*Pachitariu et al., 2023*; *Steinmetz et al., 2021*) and wrote custom MATLAB and Python code to sort waveforms into clusters arising from individual motor units (*Figure 1—figure supplement 2e–h*). Our modifications to Kilosort reflect the different challenges inherent in sorting signals from neurons recorded with

Neuropixels probes and motor units recorded with Myomatrix arrays (*Loeb and Gans, 1986*). These modifications include the following:

*Modification of spatial masking*: Individual motor units contain multiple muscle fibers (each of which is typically larger than a neuron's soma), and motor unit waveforms can often be recorded across spatially distant electrode contacts as the waveforms propagate along muscle fibers. In contrast, Kilosort – optimized for the much more local signals recorded from neurons – uses spatial masking to penalize templates that are spread widely across the electrode array. Our modifications to Kilosort therefore include ensuring that Kilosort searches for motor unit templates across all (and only) the electrode channels inserted into a given muscle. In the GitHub repository linked below, this is accomplished by setting parameter **nops.sigmaMask** to infinity, which effectively eliminates spatial masking in the analysis of the 32 unipolar channels recorded from the injectable Myomatrix array schematized in *Figure 1—figure supplement 1g*. In cases including chronic recording from mice where only a single eight-contact thread is inserted into each muscle, a similar modification can be achieved with a finite value of **nops.sigmaMask** by setting parameter **NchanNear**, which represents the number of nearby EMG channels to be included in each cluster, to equal the number of unipolar or bipolar data channels recorded from each thread. Finally, note that in all cases Kilosort parameter **NchanNearUp** (which defines the maximum number of channels across which spike templates can appear) must be reset to be equal to or less than the total number of Myomatrix data channels.

*Allowing more complex spike waveforms*: We also modified Kilosort to account for the greater duration and complexity (relative to neural spikes) of many motor unit waveforms. In the code repository linked below, Kilosort 2.5 was modified to allow longer spike templates (151 samples instead of 61), more spatiotemporal PCs for spikes (12 instead of 6), and more left/right eigenvector pairs for spike template construction (6 pairs instead of 3) to account for the greater complexity and longer duration of motor unit action potentials (*Loeb and Gans, 1986*) compared to the neural action potentials for which Kilosort was initially created. These modifications were crucial for improving sorting performance in the nonhuman primate dataset shown in *Figure 3*, and in a subset of the rodent datasets (although they were not used in the analysis of mouse data shown in *Figure 1* and *Figure 1—figure supplement 2a–f*).

Individual motor units were identified from 'candidate' units by assessing motor unit waveform consistency, SNR, and spike count, inspecting autocorrelograms to ensure that each identified units displayed an absolute refractory period of less than 1 ms, and examining cross-correlograms with other sorted units to ensure that each motor unit's waveforms were being captured by only one candidate unit. Candidate units with inconsistent waveforms or >1% of inter-spike intervals above 1 ms were discarded. Candidate units with highly similar waveform shapes and cross-correlation peaks at lag zero were merged, resulting in sorted units with well-differentiated waveform shapes and firing patterns (*Figure 1—figure supplement 2e and f*). Our spike-sorting code, which includes the above-mentioned modifications to Kilosort, is available at https://github.com/JonathanAMichaels/PixelProcessingPipeline/releases/tag/v0.9 (copy archived at *Michaels, 2023*).

Our approach to spike sorting shares the same ultimate goal as prior work using skin-surface electrode arrays to isolate signals from individual motor units but pursues this goal using different hardware and analysis approaches. A number of groups have developed algorithms for reconstructing the spatial location and spike times of active motor units (*Negro et al., 2016*; *van den Doel et al., 2008*) based on skin-surface recordings, in many cases drawing inspiration from earlier efforts to localize cortical activity using EEG recordings from the scalp (*Michel et al., 2004*). Our approach differs substantially. In Myomatrix arrays, the close electrode spacing and very close proximity of the contacts to muscle fibers ensure that each Myomatrix channel records from a much smaller volume of tissue than skin-surface arrays. This difference in recording volume in turn creates different challenges for motor unit isolation: compared to skin-surface recordings, Myomatrix recordings include a smaller number of motor units represented on each recording channel, with individual motor units appearing on a smaller fraction of the sensors than typical in a skin-surface recording. Because of this sensor-dependent difference in motor unit source mixing, different analysis approaches are required for each type of dataset. Specifically, skin-surface EMG analysis methods typically use source-separation approaches that assume that each sensor receives input from most or all of the individual sources

within the muscle as is presumably the case in the data. In contrast, the much sparser recordings from Myomatrix are better decomposed using methods like Kilosort, which are designed to extract waveforms that appear only on a small, spatially restricted subset of recording channels.

### Additional recording methods: Mouse forelimb muscle

All procedures described below were approved by the Institutional Animal Care and Use Committee at Emory University (IACUC protocol #201700359; data in *Figure 1c*, *Figure 1—figure supplement 3*) or were carried out in accordance with the European Union Directive 86/609/EEC and approved by the Champalimaud Centre for the Unknown Ethics Committee and the Portuguese Direção Geral de Veterinária (ref. no. 0421/000/000/2020; data in *Figure 1b*, *Figure 1—figure supplement 2e*). Individual Myomatrix threads were implanted in the triceps muscle using the 'intramuscular' method described above under isoflurane anesthesia (1–4% at flow rate 1 L/min). EMG data were then recorded either during home cage exploration or while animals walked on a custom-built linear treadmill (*Darmohray et al., 2019*) at speeds ranging from 15 to 25 cm/s. A 45° angled mirror below the treadmill allowed simultaneous side and bottom views of the mouse (*Machado et al., 2015*) using a single monochrome usb3 camera (Grasshopper3, Teledyne FLIR) to collect images 330 fps. We used DeepLabCut (*Mathis et al., 2018*) to track paw, limb, and body positions. These tracked points were used to identify the stride cycles of each limb, defining stance onset as the time at which each paw contacts the ground and swing onset as the time when each paw leaves the ground.

### Additional recording methods: Mouse orofacial muscle

All procedures described below were approved by the Institutional Animal Care and Use Committee at Johns Hopkins University (IACUC protocol #MO21M195). Individual Myomatrix threads were implanted on the masseter muscle using the 'epimysial' method described above. A ground pin was placed over the right visual cortex. As described previously (*Severson et al., 2017*), EMG signals and high-speed video of the orofacial area were recorded simultaneously in head-fixed animals under isoflurane anesthesia (0.9–1.5% at flow rate 1 L/min). During data collection, the experimenter used a thin wooden dowel to gently displace the mandible to measure both jaw displacement and muscle activity from the jaw jerk reflex. Jaw kinematics were quantified using a high-speed camera (PhotonFocus DR1-D1312-200-G2-8) at 400 fps using an angled mirror to collect side and bottom views simultaneously. Jaw displacement was quantified by tracking 11 keypoints along the jaw using DeepLabCut (*Mathis et al., 2018*).

### Additional recording methods: Rat forelimb muscle

All procedures described below were approved by the Institutional Animal Care and Use Committee at Emory University (IACUC protocol #201700525). Anesthesia was induced with an initial dose of 4% isoflurane in oxygen provided in an induction chamber with 2 L/min rate and maintained with 3% isoflurane at 1 L/min. Following this, rats received a subcutaneous injection of 1 mg/kg meloxicam, a subcutaneous injection of 1% lidocaine and topical application of lidocaine ointment (5%) at each incision site. Myomatrix threads were implanted in the triceps muscle using the 'intramuscular' method. EMG data were then recorded while animals walked on a treadmill at speeds ranging from 8 to 25 cm/s. Kinematics were quantified using a circular arrangement of four high-speed FLIR Black Fly S USB3 cameras (BFS-U3-16S2M-CS, Mono), each running at 125 fps. We used DeepLabCut to label pixel locations of each of the 10 anatomical landmarks on the limbs and body, which we then transformed into 3D Cartesian coordinates using Anipose (*Mathis et al., 2018*; *Karashchuk et al., 2021*). We then defined the onset of each swing/stance cycle by using local minima in the rat's forelimb endpoint position along the direction of locomotion.

### Additional recording methods: Rat forelimb muscle

All procedures described below were approved by the Institutional Animal Care and Use Committee at Johns Hopkins University (IACUC protocol #RA21M45). Prior to electrode implantation, rodents were trained for 4–6 wk to perform a single pellet reach task (*Whishaw et al., 1993*). Rodents were food-restricted for 17–18 hr prior to training. During the task, rats used the right arm to reach for sucrose pellets through a vertical slit (width = 1 cm) in a custom-built acrylic chamber. Individual Myomatrix threads were implanted on the right flexor digitorum profundus (FDP) muscle using the

'epimysial' method described above under 2.0–3.0% isoflurane anesthesia in oxygen gas. The arrays were then connected to data collection hardware via a flexible tether and EMG data were recorded while unrestrained animals performed the reaching task. Kinematics were quantified using a smart-phone camera running at 60 fps. Two raters then manually labeled the frames of grasp onset. Grasp initiation was defined when a frame of full-digit extension was immediately followed by a frame of digit flexion.

### Additional recording methods: Mouse pelvic muscle

All experimental procedures were carried out in accordance with the European Union Directive 86/609/EEC and approved by the Champalimaud Centre for the Unknown Ethics Committee and the Portuguese Direção Geral de Veterinária (ref. no. 0421/000/000/2022). As described in detail elsewhere (*Lenschow et al., 2022*), spinal optogenetic stimulation of the motor neurons innervating the BSM was performed on 2–3-month-old male BL6 mice that had received an injection of a rAAV-CAG-ChR2 into the BSM on postnatal days 3–6. Individual Myomatrix threads were implanted in the BSM using the 'intramuscular' method described above. During EMG recording, an optrode was moved on the top of the spinal cord along the rostral-caudal axis while applying optical stimulation pulses (10 ms duration, power 1–15 mW).

### Additional recording methods: Songbird vocal and respiratory muscles

All procedures described below were approved by the Institutional Animal Care and Use Committee at Emory University (IACUC protocol #201700359). As described previously (*Srivastava et al., 2015*; *Zia et al., 2020*; *Zia et al., 2018*), adult male Bengalese finches (>90 days old) were anesthetized using intramuscular injections of 40 mg/kg ketamine and 3 mg/kg midazolam, and anesthesia was maintained using 1–5% isoflurane in oxygen gas. To record from the expiratory (respiratory) muscles, an incision was made dorsal to the leg attachment and rostral to the pubic bone and the electrode array was placed on the muscle surface using the 'epimysial' approach described above. To record from syringeal (vocal) muscles, the vocal organ was accessed for electrode implantation via a midline incision into the intraclavicular air sac as described previously (*Srivastava et al., 2015*) to provide access to the ventral syringeal (VS) muscle located on the ventral portion of the syrinx near the midline.

### Additional recording methods: Cat soleus muscle

All procedures described below were approved by the Institutional Animal Care and Use Committee at Temple University (IACUC protocol #5113). As described previously (*Zaback et al., 2022*), an adult female cat was provided atropine (0.05 mg/kg intramuscular) and anesthetized with isoflurane (1.5–3.5% in oxygen), during which a series of surgical procedures were performed including L3 laminectomy, implantation of nerve cuffs on the tibial and sural nerve, and isolation of hindlimb muscles. Individual Myomatrix threads were implanted in hindlimb muscles using the 'intramuscular' method described above. Following these procedures, a precollicular decerebration was performed and isoflurane was discontinued. Following a recovery period, the activity of hindlimb motor units was recorded in response to electrical stimulation of either the contralateral tibial nerve or the ipsilateral sural nerve.

### Additional recording methods: Hawkmoth larva (caterpillar) body wall muscle

EMG recordings were obtained from fifth-instar larvae of the tobacco hornworm *Manduca sexta* using a semi-intact preparation called the 'flaterpillar' as described previously (Metallo, White, and Trimmer 2011). Briefly, after chilling animals on ice for 30 min, an incision was made along the cuticle, allowing the nerve cord and musculature to be exposed and pinned down in a Sylgard dish under cold saline solution. This preparation yields spontaneous muscle activity (fictive locomotion), which was recorded from the dorsal intermediate medial (DIM) muscle using the 'epimysial' method described above, with the modification that sutures were not used to hold the array in place.

### Additional recording methods: Frog hindlimb muscles

Spinal bullfrogs were prepared under anesthesia in accordance with USDA and PHS guidelines and regulations following approval from the Institutional Animal Care and Use Committee at Drexel University (IACUC protocol #LA-21-722 and #LA-21-709) as described previously (*Kim et al., 2019*).

Bullfrogs were anesthetized with 5% tricaine (MS-222, Sigma), spinalized, and decerebrated. The frog was placed on a support and Myomatrix arrays were implanted into the semimembranosus (SM) hind-limb muscle using the 'intramuscular' method described above. Epidermal electrical stimulation at the heel dorsum (500 ms train of 1 ms, 5 V biphasic pulses delivered at 40 Hz) or foot pinch was used to evoke reflexive motor activity.

### Additional recording methods: Rhesus macaque forelimb muscle

All procedures described below were approved by the Institutional Animal Care and Use Committee at Western University (IACUC protocol #2022-028). One male rhesus monkey (Monkey M, *Macaca mulatta*, 10 kg) was trained to perform a range of reaching tasks while seated in a robotic exoskeleton (NHP KINARM, Kingston, ON). As described previously (*Pruszynski et al., 2014*; *Scott, 1999*), this robotic device allows movements of the shoulder and elbow joints in the horizontal plane and can independently apply torque at both joints. Visual cues and hand feedback were projected from an LCD monitor onto a semi-silvered mirror in the horizontal plane of the task and direct vision of the arm was blocked with a physical barrier.

An injectable Myomatrix array (*Figure 1—figure supplement 1g*) was inserted percutaneously as shown in *Figure 1—figure supplement 1i*. Then, using his right arm, Monkey M performed a reaching task similar to previous work (*Pruszynski et al., 2014*). On each trial, the monkey waited with its fingertip in in a central target (located under the fingertip when the shoulder and elbow angles were 32° and 72°, respectively; size = 0.6 cm diameter) while countering a constant elbow load (–0.05 Nm). The monkey was presented with one of two peripheral goal targets (30/84° and 34/60° shoulder/elbow, 8 cm diameter), and after a variable delay (1.2–2 s) received one of two unpredictable elbow perturbations (±0.15 Nm step-torque) which served as a go cue to reach to the goal target. At the time of perturbation onset, all visual feedback was frozen until the hand remained in the goal target for 800 ms, after which a juice reward was given. In 10% of trials, no perturbation was applied, and the monkey had to maintain the hand in the central target. In addition to Myomatrix injectables, we acquired bipolar electromyographic activity from nonhuman primates using intramuscular fine-wire electrodes in the biceps brachii long head as described previously (*Maeda et al., 2021*), recording in this instance from the same biceps muscle in the same animal from which we also collected Myomatrix data, although in a separate recording session. Fine-wire electrodes were spaced ~8 mm apart and aligned to the muscle fibers, and a reference electrode was inserted subcutaneously in the animal's back. Muscle activity was recorded at 2000 Hz, zero-phase bandpass filtered (25–500 Hz, fourth-order Butterworth) and full-wave rectified.

## Acknowledgements

The authors thank the participants of the Emory-SKAN Remote Workshop for Advanced EMG Methods, which brought together over 100 researchers from around the world, for their critical feedback on how to improve and refine the electrode technology described here. Dr. Andrew Miri for helpful discussions and sharing locomotor EMG data from *Miri et al., 2017*. Dr. Cinzia Metallo for the initial *Manduca* studies using flexible electrode arrays. Drs. Gabriela J Martins and Mariana Correia for project and colony coordination. Dr. Ana Gonçalves for technical assistance in mouse locomotion experiments. Mattia Rigotti, Margo Shen, Nevin Aresh, and Manikandan Venkatesh for assistance in collecting rat forelimb data. Components of all figures were created using BioRender.com.

## Additional information

#### Competing interests

Chethan Pandarinath, Muhannad Bakir: Reviewing editor, *eLife*. The other authors declare that no competing interests exist.

## Funding

| Funder | Grant reference number | Author |
|---|---|---|
| National Institute of Neurological Disorders and Stroke | DP2NS105555 | Eiman Azim |
| National Institute of Neurological Disorders and Stroke | R01NS109237 | Bryce Chung<br>Muneeb Zia<br>Jonathan A Michaels<br>Amanda Jacob<br>Andrea Pack<br>Matthew J Williams<br>Kailash Nagapudi<br>Lay Heng Teng<br>Eduardo Arrambide<br>Logan Ouellette<br>Nicole Oey<br>Rhuna Gibbs<br>Philip Anschutz<br>Jiaao Lu<br>Yu Wu<br>Muhannad Bakir<br>Samuel J Sober |
| European Research Council | 866237 | Megan R Carey |
| Simons Foundation | Simons-Emory International Consortium on Motor Control | Abigail Person<br>Megan R Carey<br>Chethan Pandarinath<br>Rui M Costa<br>J Andrew Pruszynski<br>Samuel J Sober |
| Deutsche Forschungsgemeinschaft | SFB1451 | Graziana Gatto |
| Halle Institute for Global Research, Emory University | | Graziana Gatto<br>Samuel J Sober |
| National Institute of Neurological Disorders and Stroke | R01NS104194 | Taegyo Kim<br>Simon Giszter |
| National Institute of Neurological Disorders and Stroke | R01NS112959 | Graziana Gatto<br>Martyn Goulding |
| Human Frontier Science Program | LT000353/2018-L4 | Constanze Lenschow |
| European Regional Development Fund | PORTUGAL 2020 Partnership Agreement | Susana Q Lima |
| Fundação para a Ciência e a Tecnologia | Project LISBOA-01-0145-FEDER-022170 | Susana Q Lima |
| European Research Council | 772827 | Susana Q Lima |
| Fundação para a Ciência e a Tecnologia | PhD Fellowship PD/BD141576/2018 | Ana Rita Mendes |
| Banting Research Foundation | | Jonathan A Michaels |
| Canada First Research Excellence Fund | | J Andrew Pruszynski |
| Vector Institute | | Jonathan A Michaels |
| Swiss National Science Foundation | P400PM_183904 | Alice C Mosberger |

| Funder | Grant reference number | Author |
|---|---|---|
| National Institute of Neurological Disorders and Stroke | K99NS126307 | Alice C Mosberger |
| National Institute of Neurological Disorders and Stroke | R01NS104834 | Daniel O'Connor |
| National Institute of Mental Health | F32MH120873 | Jeong Jun Kim |
| Defense Advanced Research Projects Agency | PA-18-02-04-INI-FP-021 | Chethan Pandarinath |
| National Science Foundation | NCS 1835364 | Chethan Pandarinath |
| National Institutes of Health | K12HD073945 | Chethan Pandarinath |
| Alfred P. Sloan Foundation | | Chethan Pandarinath |
| Canadian Institutes of Health Research | PJT-175010 | J Andrew Pruszynski |
| Canada Research Chairs | | J Andrew Pruszynski |
| Natural Sciences and Engineering Research Council of Canada | RGPIN-2022-04421 | J Andrew Pruszynski |
| National Institute of Neurological Disorders and Stroke | R01NS084844 | Samuel J Sober |
| National Institute of Biomedical Imaging and Bioengineering | R01EB022872 | Samuel J Sober |
| National Science Foundation | 1822677 | Kyle A Thomas Andrea Pack Samuel J Sober |
| McKnight Foundation | | Muhannad Bakir Samuel J Sober |
| Kavli Foundation | | Samuel J Sober |
| Azrieli Foundation | | J Andrew Pruszynski Samuel J Sober |
| Novo Nordisk Fonden | | Samuel J Sober |
| National Science Foundation | DGE-2139757 | Kiara N Quinn Nitish Thakor |
| Johns Hopkins | Johns Hopkins Discovery Award | Nitish Thakor |
| Salk Institute | Salk Alumni Fellowship Award | Ayesha Thanawalla |
| National Institute of Neurological Disorders and Stroke | R01NS124820 | Christopher K Thompson |
| National Institute of Neurological Disorders and Stroke | R34NS118412 | Barry Trimmer |
| National Science Foundation | 1050908 | Barry Trimmer |
| National Institute of Neurological Disorders and Stroke | R01NS111479 | Eiman Azim |

| Funder | Grant reference number | Author |
|---|---|---|
| National Institute of Neurological Disorders and Stroke | RF1NS128898 | Eiman Azim |
| National Institute of Neurological Disorders and Stroke | U19NS112959 | Eiman Azim |
| National Institute of Neurological Disorders and Stroke | U24NS126936 | Bryce Chung<br>Muneeb Zia<br>Jonathan A Michaels<br>Amanda Jacob<br>Andrea Pack<br>Matthew J Williams<br>Kailash Nagapudi<br>Lay Heng Teng<br>Eduardo Arrambide<br>Logan Ouellette<br>Nicole Oey<br>Rhuna Gibbs<br>Philip Anschutz<br>Jiaao Lu<br>Yu Wu<br>Muhannad Bakir<br>Samuel J Sober |
| National Institute of Neurological Disorders and Stroke | F31NS124347 | Simon Giszter<br>Taegyo Kim |
| National Institute of Neurological Disorders and Stroke | R01NS111643 | Graziana Gatto<br>Martyn Goulding |
| Swiss National Science Foundation | P2EZP3_172128 | Alice C Mosberger |
| National Institutes of Health | DP2NS127291 | Chethan Pandarinath |
| National Science Foundation | DGE-1937971 | Kyle A Thomas |
| National Science Foundation | DGE-1444932 | Andrea Pack |

The funders had no role in study design, data collection and interpretation, or the decision to submit the work for publication.

## Author contributions

Bryce Chung, Conceptualization, Data curation, Software, Formal analysis, Supervision, Validation, Investigation, Visualization, Methodology, Writing – original draft, Project administration, Writing – review and editing; Muneeb Zia, Conceptualization, Software, Formal analysis, Supervision, Investigation, Visualization, Methodology, Writing – original draft, Project administration, Writing – review and editing; Kyle A Thomas, Conceptualization, Data curation, Software, Formal analysis, Validation, Investigation, Visualization, Methodology, Writing – original draft, Writing – review and editing; Jonathan A Michaels, J Andrew Pruszynski, Conceptualization, Data curation, Software, Formal analysis, Funding acquisition, Validation, Investigation, Visualization, Methodology, Writing – original draft, Writing – review and editing; Amanda Jacob, Conceptualization, Supervision, Investigation, Methodology, Writing – original draft, Project administration, Writing – review and editing; Andrea Pack, Kailash Nagapudi, Jiaao Lu, Conceptualization, Investigation, Methodology; Matthew J Williams, Conceptualization, Software, Formal analysis, Validation, Investigation, Methodology, Writing – original draft; Lay Heng Teng, Eduardo Arrambide, Logan Ouellette, Nicole Oey, Rhuna Gibbs, Philip Anschutz, Yu Wu, Mehrdad Kashefi, Tomomichi Oya, Rhonda Kersten, Runming Wang, William Olson, Kiara N Quinn, Pierce Perkins, Susan Coltman, Trevor Smith, Ben Binder-Markey, Martin Zaback, Investigation, Methodology; Alice C Mosberger, Christopher K Thompson, Martyn Goulding, Nitish Thakor, Funding acquisition, Investigation, Methodology; Sean O'Connell, Data curation, Software, Formal

analysis, Validation, Investigation, Visualization, Methodology, Writing – original draft; Hugo Marques, Data curation, Investigation, Methodology, Writing – original draft; Ana Rita Mendes, Data curation, Funding acquisition, Investigation, Methodology, Writing – original draft; Constanze Lenschow, Simon Giszter, Eiman Azim, Barry Trimmer, Susana Q Lima, Funding acquisition, Investigation, Methodology, Writing – original draft; Gayathri Kondakath, Taegyo Kim, Investigation, Methodology, Writing – original draft; Jeong Jun Kim, Software, Formal analysis, Investigation, Visualization, Methodology, Writing – original draft; Graziana Gatto, Ayesha Thanawalla, Conceptualization, Funding acquisition, Investigation, Methodology; Abigail Person, Rui M Costa, Funding acquisition, Investigation, Methodology, Writing – original draft, Writing – review and editing; Daniel O'Connor, Software, Formal analysis, Funding acquisition, Investigation, Methodology, Writing – original draft; Megan R Carey, Data curation, Funding acquisition, Investigation, Methodology, Writing – original draft, Writing – review and editing; Chethan Pandarinath, Data curation, Software, Formal analysis, Funding acquisition, Validation, Investigation, Visualization, Methodology, Writing – original draft, Writing – review and editing; Muhannad Bakir, Conceptualization, Software, Formal analysis, Supervision, Funding acquisition, Investigation, Visualization, Methodology, Writing – original draft, Writing – review and editing; Samuel J Sober, Conceptualization, Data curation, Software, Formal analysis, Supervision, Funding acquisition, Validation, Investigation, Visualization, Methodology, Writing – original draft, Writing – review and editing

#### Author ORCIDs
Jonathan A Michaels https://orcid.org/0000-0002-5179-3181
Jiaao Lu http://orcid.org/0000-0002-7973-1679
Mehrdad Kashefi http://orcid.org/0000-0001-5981-5923
Tomomichi Oya http://orcid.org/0000-0003-1233-6356
Alice C Mosberger https://orcid.org/0000-0003-1114-1469
Hugo Marques https://orcid.org/0000-0002-8709-4841
Graziana Gatto https://orcid.org/0000-0002-4244-8925
Ben Binder-Markey https://orcid.org/0000-0001-8920-4381
Abigail Person https://orcid.org/0000-0001-9805-7600
Eiman Azim http://orcid.org/0000-0002-1015-1772
Megan R Carey http://orcid.org/0000-0002-4499-1657
Rui M Costa http://orcid.org/0000-0003-1707-1051
J Andrew Pruszynski http://orcid.org/0000-0003-0786-0081
Samuel J Sober http://orcid.org/0000-0002-1140-7469

#### Ethics
All procedures described in this study were approved by the appropriate Institutional Animal Care and Use Committee (at Emory University, Johns Hopkins University, Temple University, Drexel University, the Champalimaud Neuroscience Program, or Western University).

Reviewer #1 (Public Review): https://doi.org/10.7554/eLife.88551.3.sa1
Reviewer #2 (Public Review): https://doi.org/10.7554/eLife.88551.3.sa2
Author Response https://doi.org/10.7554/eLife.88551.3.sa3

---

# Additional files

#### Supplementary files
• MDAR checklist

#### 12345678910111Data availability
A data archive including two EMG datasets recorded with Myomatrix arrays from behaving animals is available at https://doi.org/10.5061/dryad.66t1g1k70. This archive includes the mouse triceps data shown in *Figure 1 (b and d)* and *Figure 1—figure supplement 2a, b, e, and f*, the rhesus macaque biceps data shown in *Figure 3* and *Figure 1—figure supplement 2g*, and associated metadata. Our spike -sorting code is available at https://github.com/JonathanAMichaels/PixelProcessingPipeline (copy archived at *Michaels, 2023*).

The following dataset was generated:

| Author(s) | Year | Dataset title | Dataset URL | Database and Identifier |
|---|---|---|---|---|
| Chung, Bryce et al. | 2023 | Myomatrix arrays for high-definition muscle recording | https://doi.org/10.5061/dryad.66t1g1k70 | Dryad Digital Repository, 10.5061/dryad.66t1g1k70 |

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
