## [Editor Report · eLife assessment]

This **important** paper reports major technical advances for in vivo intramuscular electrical recording from multiple motor units in behaving animals. The paper includes **compelling** demonstrations of the efficacy of this new technique in multiple animal species. This new muscle recording method has the potential to provide new insight into a wide range of questions in motor neuroscience.

---

## [Referee Report · Reviewer #1 (Public Review)]

Motoneurons constitute the final common pathway linking central impulse traffic to behavior, and neurophysiology faces an urgent need for methods to record their activity at high resolution and scale in intact animals during natural movement. In this consortium manuscript, Chung et al. introduce high-density electrode arrays on a flexible substrate that can be implanted into muscle, enabling the isolation of multiple motor units during movement. They then demonstrate these arrays can produce high-quality recordings in a wide range of species, muscles, and tasks. The methods are explained clearly, and the claims are justified by the data. While technical details on the arrays have been published previously, the main significance of this manuscript is the application of this new technology to different muscles and animal species during naturalistic behaviors. Overall, we feel the manuscript will be of significant interest to researchers in motor systems and muscle physiology.

The authors have thoroughly addressed all our original comments, and we have no further concerns.

---

## [Referee Report · Reviewer #2 (Public Review)]

This work provides a novel design of implantable and high-density EMG electrodes to study muscle physiology and neuromotor control at the level of individual motor units. Current methods of recording EMG using intramuscular fine-wire electrodes do not allow for isolation of motor units and are limited by the muscle size and the type of behavior used in the study. The authors of myomatrix arrays had set out to overcome these challenges in EMG recording and provided compelling evidence to support the usefulness of the new technology.

Strengths:

• They presented convincing examples of EMG recordings with high signal quality using this new technology from a wide array of animal species, muscles, and behavior.

• The design included suture holes and pull-on tabs that facilitate implantation and ensure stable recordings over months.

• Clear presentation of specifics of the fabrication and implantation, recording methods used, and data analysis

I am satisfied with the authors' response to my previous concerns on the weaknesses of the study.

---

## [Author Response]

The following is the authors’ response to the original reviews.

We thank all three Reviewers for their comments and have revised the manuscript accordingly.

**Reviewer #1 (Public Review):**
The main objective of this paper is to report the development of a new intramuscular probe that the authors have named Myomatrix arrays. The goal of the Myomatrix probe is to significantly advance the current technological ability to record the motor output of the nervous system, namely fine-wire electromyography (EMG). Myomatrix arrays aim to provide large-scale recordings of multiple motor units in awake animals under dynamic conditions without undue movement artifacts and maintain long-term stability of chronically implanted probes. Animal motor behavior occurs through muscle contraction, and the ultimate neural output in vertebrates is at the scale of motor units, which are bundles of muscle fibers (muscle cells) that are innervated by a single motor neuron. The authors have combined multiple advanced manufacturing techniques, including lithography, to fabricate large and dense electrode arrays with mechanical features such as barbs and suture methods that would stabilize the probe's location within the muscle without creating undue wiring burden or tissue trauma. Importantly, the fabrication process they have developed allows for rapid iteration from design conception to a physical device, which allows for design optimization of the probes for specific muscle locations and organisms. The electrical output of these arrays is processed through a variety of means to try to identify single motor unit activity. At the simplest, the approach is to use thresholds to identify motor unit activity. Of intermediate data analysis complexity is the use of principal component analysis (PCA, a linear second-order regression technique) to disambiguate individual motor units from the wide field recordings of the arrays, which benefits from the density and numerous recording electrodes. At the highest complexity, they use spike sorting techniques that were developed for Neuropixels, a large-scale electrophysiology probe for cortical neural recordings. Specifically, they use an estimation code called kilosort, which ultimately relies on clustering techniques to separate the multi-electrode recordings into individual spike waveforms.The biggest strength of this work is the design and implementation of the hardware technology. It is undoubtedly a major leap forward in our ability to record the electrical activity of motor units. The myomatrix arrays trounce fine-wire EMGs when it comes to the quality of recordings, the number of simultaneous channels that can be recorded, their long-term stability, and resistance to movement artifacts.The primary weakness of this work is its reliance on kilosort in circumstances where most of the channels end up picking up the signal from multiple motor units. As the authors quite convincingly show, this setting is a major weakness for fine-wire EMG. They argue that the myomatrix array succeeds in isolating individual motor unit waveforms even in that challenging setting through the application of kilosort.Although the authors call the estimated signals as well-isolated waveforms, there is no independent evidence of the accuracy of the spike sorting algorithm. The additional step (spike sorting algorithms like kilosort) to estimate individual motor unit spikes is the part of the work in question. Although the estimation algorithms may be standard practice, the large number of heuristic parameters associated with the estimation procedure are currently tuned for cortical recordings to estimate neural spikes. Even within the limited context of Neuropixels, for which kilosort has been extensively tested, basic questions like issues of observability, linear or nonlinear, remain open. By observability, I mean in the mathematical sense of well-posedness or conditioning of the inverse problem of estimating single motor unit spikes given multi-channel recordings of the summation of multiple motor units. This disambiguation is not always possible. kilosort's validation relies on a forward simulation of the spike field generation, which is then truth-tested against the sorting algorithm. The empirical evidence is that kilosort does better than other algorithms for the test simulations that were performed in the context of cortical recordings using the Neuropixels probe. But this work has adopted kilosort without comparable truth-tests to build some confidence in the application of kilosort with myomatrix arrays.

Kilosort was developed to analyze spikes from neurons rather than motor units and, as Reviewer #1 correctly points out, despite a number of prior validation studies the conditions under which Kilosort accurately identifies individual neurons are still incompletely understood. Our application of Kilosort to motor unit data therefore demands that we explain which of Kilosort’s assumptions do and do not hold for motor unit data and explain how our modifications of the Kilosort pipeline to account for important differences between neural and muscle recording, which we summarize below and have included in the revised manuscript.

Additionally, both here and in the revised paper we emphasize that while the presented spike sorting methods (thresholding, PCA-based clustering, and Kilosort) robustly extract motor unit waveforms, spike sorting of motor units is still an ongoing project. Our future work will further elaborate how differences between cortical and motor unit data should inform approaches to spike sorting as well as develop simulated motor unit datasets that can be used to benchmark spike sorting methods.

For our current revision, we have added detailed discussion (see “Data analysis: spike sorting”) of the risks and benefits of our use of Kilosort to analyze motor unit data, in each case clarifying how we have modified the Kilosort code with these issues in mind:

“Modification of spatial masking: Individual motor units contain multiple muscle fibers (each of which is typically larger than a neuron’s soma), and motor unit waveforms can often be recorded across spatially distant electrode contacts as the waveforms propagate along muscle fibers. In contrast, Kilosort - optimized for the much more local signals recorded from neurons - uses spatial masking to penalize templates that are spread widely across the electrode array. Our modifications to Kilosort therefore include ensuring that Kilosort search for motor unit templates across all (and only) the electrode channels inserted into a given muscle. In this Github repository linked above, this is accomplished by setting parameter nops.sigmaMask to infinity, which effectively eliminates spatial masking in the analysis of the 32 unipolar channels recorded from the injectable Myomatrix array schematized in Supplemental Figure 1g. In cases including chronic recording from mice where only a single 8-contact thread is inserted into each muscle, a similar modification can be achieved with a finite value of nops.sigmaMask by setting parameter NchanNear, which represents the number of nearby EMG channels to be included in each cluster, to equal the number of unipolar or bipolar data channels recorded from each thread. Finally, note that in all cases Kilosort parameter NchanNearUp (which defines the maximum number of channels across which spike templates can appear) must be reset to be equal to or less than the total number of Myomatrix data channels.”

“Allowing more complex spike waveforms: We also modified Kilosort to account for the greater duration and complexity (relative to neural spikes) of many motor unit waveforms. In the code repository linked above, Kilosort 2.5 was modified to allow longer spike templates (151 samples instead of 61), more spatiotemporal PCs for spikes (12 instead of 6), and more left/right eigenvector pairs for spike template construction (6 pairs instead of 3). These modifications were crucial for improving sorting performance in the nonhuman primate dataset shown in Figure 3, and in a subset of the rodent datasets (although they were not used in the analysis of mouse data shown in Fig. 1 and Supplemental Fig. 2a-f).”

Furthermore, as the paper on the latest version of kilosort, namely v4, discusses, differences in the clustering algorithm is the likely reason for kilosort4 performing more robustly than kilosort2.5 (used in the myomatrix paper). Given such dependence on details of the implementation and the use of an older kilosort version in this paper, the evidence that the myomatrix arrays truly record individual motor units under all the types of data obtained is under question.

We chose to modify Kilosort 2.5, which has been used by many research groups to sort spike features, rather than the just-released Kilosort 4.0. Although future studies might directly compare the performance of these two versions on sorting motor unit data, we feel that such an analysis is beyond the scope of this paper, which aims primarily to introduce our electrode technology and demonstrate that a wide range of sorting methods (thresholding, PCA-based waveform clustering, and Kilosort) can all be used to extract single motor units. Additionally, note that because we have made several significant modifications to Kilosort 2.5 as described above, it is not clear what a “direct” comparison between different Kilosort versions would mean, since the procedures we provide here are no longer identical to version 2.5.

There is an older paper with a similar goal to use multi-channel recording to perform sourcelocalization that the authors have failed to discuss. Given the striking similarity of goals and the divergence of approaches (the older paper uses a surface electrode array), it is important to know the relationship of the myomatrix array to the previous work. Like myomatrix arrays, the previous work also derives inspiration from cortical recordings, in that case it uses the approach of source localization in large-scale EEG recordings using skull caps, but applies it to surface EMG arrays. Ref: van den Doel, K., Ascher, U. M., & Pai, D. K. (2008). Computed myography: three-dimensional reconstruction of motor functions from surface EMG data. Inverse Problems, 24(6), 065010.

We thank the Reviewer for pointing out this important prior work, which we now cite and discuss in the revised manuscript under “Data analysis: spike sorting” [lines 318-333]:

“Our approach to spike sorting shares the same ultimate goal as prior work using skin-surface electrode arrays to isolate signals from individual motor units but pursues this goal using different hardware and analysis approaches. A number of groups have developed algorithms for reconstructing the spatial location and spike times of active motor units (Negro et al. 2016; van den Doel, Ascher, and Pai 2008) based on skin-surface recordings, in many cases drawing inspiration from earlier efforts to localize cortical activity using EEG recordings from the scalp (Michel et al. 2004). Our approach differs substantially. In Myomatrix arrays, the close electrode spacing and very close proximity of the contacts to muscle fibers ensure that each Myomatrix channel records from a much smaller volume of tissue than skin-surface arrays. This difference in recording volume in turn creates different challenges for motor unit isolation: compared to skin-surface recordings, Myomatrix recordings include a smaller number of motor units represented on each recording channel, with individual motor units appearing on a smaller fraction of the sensors than typical in a skin-surface recording. Because of this sensordependent difference in motor unit source mixing, different analysis approaches are required for each type of dataset. Specifically, skin-surface EMG analysis methods typically use source-separation approaches that assume that each sensor receives input from most or all of the individual sources within the muscle as is presumably the case in the data. In contrast, the much sparser recordings fromMyomatrix are better decomposed using methods like Kilosort, which are designed to extract waveforms that appear only on a small, spatially-restricted subset of recording channels.”

The incompleteness of the evidence that the myomatrix array truly measures individual motor units is limited to the setting where multiple motor units have similar magnitude of signal in most of the channels. In the simpler data setting where one motor dominates in some channel (this seems to occur with some regularity), the myomatrix array is a major advance in our ability to understand the motor output of the nervous system. The paper is a trove of innovations in manufacturing technique, array design, suture and other fixation devices for long-term signal stability, and customization for different muscle sizes, locations, and organisms. The technology presented here is likely to achieve rapid adoption in multiple groups that study motor behavior, and would probably lead to new insights into the spatiotemporal distribution of the motor output under more naturally behaving animals than is the current state of the field.

We thank the Reviewer for this positive evaluation and for the critical comments above.

**Reviewer #2 (Public Review):**
Motoneurons constitute the final common pathway linking central impulse traffic to behavior, and neurophysiology faces an urgent need for methods to record their activity at high resolution and scale in intact animals during natural movement. In this consortium manuscript, Chung et al. introduce highdensity electrode arrays on a flexible substrate that can be implanted into muscle, enabling the isolation of multiple motor units during movement. They then demonstrate these arrays can produce high-quality recordings in a wide range of species, muscles, and tasks. The methods are explained clearly, and the claims are justified by the data. While technical details on the arrays have been published previously, the main significance of this manuscript is the application of this new technology to different muscles and animal species during naturalistic behaviors. Overall, we feel the manuscript will be of significant interest to researchers in motor systems and muscle physiology, and we have no major concerns. A few minor suggestions for improving the manuscript follow.

We thank the Reviewer for this positive overall assessment.

The authors perhaps understate what has been achieved with classical methods. To further clarify the novelty of this study, they should survey previous approaches for recording from motor units during active movement. For example, Pflüger & Burrows (J. Exp. Biol. 1978) recorded from motor units in the tibial muscles of locusts during jumping, kicking, and swimming. In humans, Grimby (J. Physiol. 1984) recorded from motor units in toe extensors during walking, though these experiments were most successful in reinnervated units following a lesion. In addition, the authors might briefly mention previous approaches for recording directly from motoneurons in awake animals (e.g., Robinson, J. Neurophys. 1970; Hoffer et al., Science 1981).

We agree and have revised the manuscript to discuss these and other prior use of traditional EMG, including here [lines 164-167]:

“The diversity of applications presented here demonstrates that Myomatrix arrays can obtain highresolution EMG recordings across muscle groups, species, and experimental conditions including spontaneous behavior, reflexive movements, and stimulation-evoked muscle contractions. Although this resolution has previously been achieved in moving subjects by directly recording from motor neuron cell bodies in vertebrates (Hoffer et al. 1981; Robinson 1970; Hyngstrom et al. 2007) and by using fine-wire electrodes in moving insects (Pfluger 1978; Putney et al. 2023), both methods are extremely challenging and can only target a small subset of species and motor unit populations. Exploring additional muscle groups and model systems with Myomatrix arrays will allow new lines of investigation into how the nervous system executes skilled behaviors and coordinates the populations of motor units both within and across individual muscles…

For chronic preparations, additional data and discussion of the signal quality over time would be useful. Can units typically be discriminated for a day or two, a week or two, or longer?A related issue is whether the same units can be tracked over multiple sessions and days; this will be of particular significance for studies of adaptation and learning.

Although the yields of single units are greatest in the 1-2 weeks immediately following implantation, in chronic preparations we have obtained well-isolated single units up to 65 days post-implant. Anecdotally, in our chronic mouse implants we occasionally see motor units on the same channel across multiple days with similar waveform shapes and patterns of behavior-locked activity. However, because data collection for this manuscript was not optimized to answer this question, we are unable to verify whether these observations actually reflect cross-session tracking of individual motor units. For example, in all cases animals were disconnected from data collection hardware in between recording sessions (which were often separated by multiple intervening days) preventing us from continuously tracking motor units across long timescales. We agree with the reviewer that long-term motor unit tracking would be extremely useful as a tool for examining learning and plan to address this question in future studies.

We have added a discussion of these issues to the revised manuscript [lines 52-59]:

“…These methods allow the user to record simultaneously from ensembles of single motor units (Fig. 1c,d) in freely behaving animals, even from small muscles including the lateral head of the triceps muscle in mice (approximately 9 mm in length with a mass of 0.02 g 23). Myomatrix recordings isolated single motor units for extended periods (greater than two months, Supp. Fig. 3e), although highest unit yield was typically observed in the first 1-2 weeks after chronic implantation. Because recording sessions from individual animals were often separated by several days during which animals were disconnected from data collection equipment, we are unable to assess based on the present data whether the same motor units can be recorded over multiple days.”

Moreover, we have revised Supplemental Figure 3 to show an example of single motor units recorded >2 months after implantation:

**Author response image 1. sa3fig1:** Longevity of Myomatrix recordingsIn addition to isolating individual motor units, Myomatrix arrays also provide stable multi-unit recordings of comparable or superior quality to conventional fine wire EMG…. (e) Although individual motor units were most frequently recorded in the first two weeks of chronic recordings (see main text), Myomatrix arrays also isolate individual motor units after much longer periods of chronic implantation, as shown here where spikes from two individual motor units (colored boxes in bottom trace) were isolated during locomotion 65 days after implantation. This bipolar recording was collected from the subject plotted with unfilled black symbols in panel (d).

It appears both single-ended and differential amplification were used. The authors should clarify in the Methods which mode was used in each figure panel, and should discuss the advantages and disadvantages of each in terms of SNR, stability, and yield, along with any other practical considerations.

We thank the reviewer for the suggestion and have added text to all figure legends clarifying whether each recording was unipolar or bipolar.

Is there likely to be a motor unit size bias based on muscle depth, pennation angle, etc.?

Although such biases are certainly possible, the data presented here are not well-suited to answering these questions. For chronic implants in small animals, the target muscles (e.g. triceps in mice) are so small that the surgeon often has little choice about the site and angle of array insertion, preventing a systematic analysis of this question. For acute array injections in larger animals such as rhesus macaques, we did not quantify the precise orientation of the arrays (e.g. with ultrasound imaging) or the muscle fibers themselves, again preventing us from drawing strong conclusions on this topic. This question is likely best addressed in acute experiments performed on larger muscles, in which the relative orientations of array threads and muscle fibers can be precisely imaged and systematically varied to address this important issue.

Can muscle fiber conduction velocity be estimated with the arrays?

We sometimes observe fiber conduction delays up to 0.5 msec as the spike from a single motor unit moves from electrode contact to electrode contact, so spike velocity could be easily estimated given the known spatial separation between electrode contacts. However (closely related to the above question) this will only provide an accurate estimate of muscle fiber conduction velocity if the electrode contacts are arranged parallel to fiber direction, which is difficult to assess in our current dataset. If the arrays are not parallel, this computation will produce an overestimate of conduction velocity, as in the extreme case where a line of electrode contacts arranged perpendicular to the fiber direction might have identical spike arrival times, and therefore appear to have an infinite conduction velocity. Therefore, although Myomatrix arrays can certainly be used to estimate conduction velocity, such estimates should be performed in future studies only in settings where the relative orientation of array threads and muscle fibers can be accurately measured.

The authors suggest their device may have applications in the diagnosis of motor pathologies. Currently, concentric needle EMG to record from multiple motor units is the standard clinical method, and they may wish to elaborate on how surgical implantation of the new array might provide additional information for diagnosis while minimizing risk to patients.

We thank the reviewer for the suggestion and have modified the manuscript’s final paragraph accordingly [lines 182-188]:

“Applying Myomatrix technology to human motor unit recordings, particularly by using the minimally invasive injectable designs shown in Figure 3 and Supplemental Figure 1g,i, will create novel opportunities to diagnose motor pathologies and quantify the effects of therapeutic interventions in restoring motor function. Moreover, because Myomatrix arrays are far more flexible than the rigid needles commonly used to record clinical EMG, our technology might significantly reduce the risk and discomfort of such procedures while also greatly increasing the accuracy with which human motor function can be quantified. This expansion of access to high-resolution EMG signals – across muscles, species, and behaviors – is the chief impact of the Myomatrix project.”

**Reviewer #3 (Public Review):**
This work provides a novel design of implantable and high-density EMG electrodes to study muscle physiology and neuromotor control at the level of individual motor units. Current methods of recording EMG using intramuscular fine-wire electrodes do not allow for isolation of motor units and are limited by the muscle size and the type of behavior used in the study. The authors of Myomatrix arrays had set out to overcome these challenges in EMG recording and provided compelling evidence to support the usefulness of the new technology.Strengths:They presented convincing examples of EMG recordings with high signal quality using this new technology from a wide array of animal species, muscles, and behavior.• The design included suture holes and pull-on tabs that facilitate implantation and ensure stablerecordings over months.• Clear presentation of specifics of the fabrication and implantation, recording methods used, and data analysis.

We thank the Reviewer for these comments.

Weaknesses:The justification for the need to study the activity of isolated motor units is underdeveloped. The study could be strengthened by providing example recordings from studies that try to answer questions where isolation of motor unit activity is most critical. For example, there is immense value for understanding muscles with smaller innervation ratio which tend to have many motor neurons for fine control of eyes and hand muscles.

We thank the Reviewer for the suggestion and have modified the manuscript accordingly [lines 170-174]:

“…how the nervous system executes skilled behaviors and coordinates the populations of motor units both within and across individual muscles. These approaches will be particularly valuable in muscles in which each motor neuron controls a very small number of muscle fibers, allowing fine control of oculomotor muscles in mammals as well as vocal muscles in songbirds (Fig. 2g), in which most individual motor neurons innervate only 1-3 muscle fibers (Adam et al. 2021).”

**Reviewer #1 (Recommendations for The Authors):**
I would urge the authors to consider a thorough validation of the spike sorting piece of the workflow. Barring that weakness, this paper has the potential to transform motor neuroscience. The validation efforts of kilosort in the context of Neuropixels might offer a template for how to convince the community of the accuracy of myomatrix arrays in disambiguating individual motor unit waveforms.I have a few minor detailed comments, that the authors may find of some use. My overall comment is to commend the authors for the precision of the work as well as the writing. However, exercising caution associated with kilosort could truly elevate the paper by showing where there is room for improvement.

We thank the Reviewer for these comments - please see our summary of our revisions related to Kilosort in our reply to the public reviews above.

L6-7: The relationship between motor unit action potential and the force produced is quite complicated in muscle. For example, recent work has shown how decoupled the force and EMG can be during nonsteady locomotion. Therefore, it is not a fully justified claim that recording motor unit potentials will tell us what forces are produced. This point relates to another claim made by the authors (correctly) that EMG provides better quality information about muscle motor output in isometric settings than in more dynamic behaviors. That same problem could also apply to motor unit recordings and their relationship to muscle force. The relationship is undoubtedly strong in an isometric setting. But as has been repeatedly established, the electrical activity of muscle is only loosely related to its force output and lacks in predictive power.

This is an excellent point, and our revised manuscript now addresses this issue [lines 174-176]:

“…Of further interest will be combining high-resolution EMG with precise measurement of muscle length and force output to untangle the complex relationship between neural control, body kinematics, and muscle force that characterizes dynamic motor behavior. Similarly, combining Myomatrix recordings with high-density brain recordings….”

L12: There is older work that uses an array of skin mounted EMG electrodes to solve a source location problem, and thus come quite close to the authors' stated goals. However, the authors have failed to cite or provide an in-depth analysis and discussion of this older work.

As described above in the response to Reviewer 1’s public review comments, we now cite and discuss these papers.

L18-19: "These limitations have impeded our understanding of fundamental questions in motor control, ..." There are two independently true statements here. First is that there are limitations to EMG based inference of motor unit activity. Second is that there are gaps in the current understanding of motor unit recruitment patterns and modification of these patterns during motor learning. But the way the first few paragraphs have been worded makes it seem like motor unit recordings is a panacea for these gaps in our knowledge. That is not the case for many reasons, including key gaps in our understanding of how muscle's electrical activity relates to its force, how force relates to movement, and how control goals map to specific movement patterns. This manuscript would in fact be strengthened by acknowledging and discussing the broader scope of gaps in our understanding, and thus more precisely pinpointing the specific scientific knowledge that would be gained from the application of myomatrix arrays.

We agree and have revised the manuscript to note this complexity (see our reply to this Reviewer’s other comment about muscle force, above).

L140-143: The estimation algorithms yields potential spikes but lacking the validation of the sorting algorithms, it is not justifiable to conclude that the myomatrix arrays have already provided information about individual motor units.

Please see our replies to Reviewer #1s public comments (above) regarding motor unit spike sorting.

L181-182: "These methods allow very fine pitch escape routing (<10 µm spacing), alignment between layers, and uniform via formation." I find this sentence hard to understand. Perhaps there is some grammatical ambiguity?

We have revised this passage as follows [lines 194-197]:

"These methods allow very fine pitch escape routing (<10 µm spacing between the thin “escape” traces connecting electrode contacts to the connector), spatial alignment between the multiple layers of polyimide and gold that constitute each device, and precise definition of “via” pathways that connect different layers of the device.”

L240: What is the rationale for choosing this frequency band for the filter?

Individual motor unit waveforms have peak energy at roughly 0.5-2.0 kHz, although units recorded at very high SNR often have voltage waveform features at higher frequencies. The high- and lowpass cutoff frequencies should reflect this, although there is nothing unique about the 350 Hz and 7,000 Hz cutoffs we describe, and in all recordings similar results can be obtained with other choices of low/high frequency cutoffs.

L527-528: There are some key differences between the electrode array design presented here and traditional fine-wire EMG in terms of features used to help with electrode stability within the muscle. A barb-like structure is formed in traditional fine-wire EMG by bending the wire outside the canula of the needle used to place it within the muscle. But when the wire is pulled out, it is common for the barb to break off and be left behind. This is because of the extreme (thin) aspect ratio of the barb in fine wire EMG and low-cycle fatigue fracture of the wire. From the schematic shown here, the barb design seems to be stubbier and thus less prone to breaking off. This raises the question of how much damage is inflicted during the pull-out and the associated level of discomfort to the animal as a result. The authors should present a more careful statement and documentation with regard to this issue.

We have updated the manuscript to highlight the ease of inserting and removing Myomatrix probes, and to clarify that in over 100 injectable insertions/removal there have been zero cases of barbs (or any other part) of the devices breaking off within the muscle [lines 241-249]:

“…Once the cannula was fully inserted, the tail was released, and the cannula slowly removed. After recording, the electrode and tail were slowly pulled out of the muscle together. Insertion and removal of injectable Myomatrix devices appeared to be comparable or superior to traditional fine-wire EMG electrodes (in which a “hook” is formed by bending back the uninsulated tip of the recording wire) in terms of both ease of injection, ease of removal of both the cannula and the array itself, and animal comfort. Moreover, in over 100 Myomatrix injections performed in rhesus macaques, there were zero cases in which Myomatrix arrays broke such that electrode material was left behind in the recorded muscle, representing a substantial improvement over traditional fine-wire approaches, in which breakage of the bent wire tip regularly occurs (Loeb and Gans 1986).”

**Reviewer #2 (Recommendations For The Authors):**
The Abstract states the device records "muscle activity at cellular resolution," which could potentially be read as a claim that single-fiber recording has been achieved. The authors might consider rewording.

The Reviewer is correct, and we have removed the word “cellular”.

The supplemental figures could perhaps be moved to the main text to aid readers who prefer to print the combined PDF file.

After finalizing the paper we will upload all main-text and supplemental figures into a single pdf on biorXiv for readers who prefer a single pdf. However, given that the supplemental figures provide more technical and detailed information than the main-text figures, for the paper on the eLife site we prefer the current eLife format in which supplemental figures are associated with individual main-text figures online.

**Reviewer #3 (Recommendations For The Authors):**
• The work could be strengthened by showing examples of simultaneous recordings from different muscles.

Although Myomatrix arrays can indeed be used to record simultaneously from multiple muscles, in this manuscript we have decided to focus on high-resolution recordings that maximize the number of recording channels and motor units obtained from a single muscle. Future work from our group with introduce larger Myomatrix arrays optimized for recording from many muscles simultaneously.

• The implantation did not include mention of testing the myomatrix array during surgery by using muscle stimulation to verify correct placement and connection.

As the Reviewer points out electrical stimulation is a valuable tool for confirming successful EMG placement. However we did not use this approach in the current study, relying instead on anatomical confirmation of muscle targeting (e.g. intrasurgical and postmortem inspection in rodents) and by implanting large, easy-totarget arm muscles (in primates) where the risk of mis-targeting is extremely low. Future studies will examine both electrical stimulation and ultrasound methods for confirming the placement of Myomatrix arrays.

References cited above

Adam, I., A. Maxwell, H. Rossler, E. B. Hansen, M. Vellema, J. Brewer, and C. P. H. Elemans. 2021. 'One-to-one innervation of vocal muscles allows precise control of birdsong', Curr Biol, 31: 3115-24 e5.

Hoffer, J. A., M. J. O'Donovan, C. A. Pratt, and G. E. Loeb. 1981. 'Discharge patterns of hindlimb motoneurons during normal cat locomotion', Science, 213: 466-7.

Hyngstrom, A. S., M. D. Johnson, J. F. Miller, and C. J. Heckman. 2007. 'Intrinsic electrical properties of spinal motoneurons vary with joint angle', Nat Neurosci, 10: 363-9.

Loeb, G. E., and C. Gans. 1986. Electromyography for Experimentalists, First edi (The University of Chicago Press: Chicago, IL).

Michel, C. M., M. M. Murray, G. Lantz, S. Gonzalez, L. Spinelli, and R. Grave de Peralta. 2004. 'EEG source imaging', Clin Neurophysiol, 115: 2195-222.

Negro, F., S. Muceli, A. M. Castronovo, A. Holobar, and D. Farina. 2016. 'Multi-channel intramuscular and surface EMG decomposition by convolutive blind source separation', J Neural Eng, 13: 026027.

Pfluger, H. J.; Burrows, M. 1978. 'Locusts use the same basic motor pattern in swimming as in jumping and kicking', Journal of experimental biology, 75: 81-93.

Putney, Joy, Tobias Niebur, Leo Wood, Rachel Conn, and Simon Sponberg. 2023. 'An information theoretic method to resolve millisecond-scale spike timing precision in a comprehensive motor program', PLOS Computational Biology, 19: e1011170.

Robinson, D. A. 1970. 'Oculomotor unit behavior in the monkey', J Neurophysiol, 33: 393-403.

van den Doel, Kees, Uri M Ascher, and Dinesh K Pai. 2008. 'Computed myography: three-dimensional reconstruction of motor functions from surface EMG data', Inverse Problems, 24: 065010.